# Cardiac Computed Tomography Evaluation of Association of Left Ventricle Disfunction and Epicardial Adipose Tissue Density in Patients with Low to Intermediate Cardiovascular Risk

**DOI:** 10.3390/medicina59020232

**Published:** 2023-01-26

**Authors:** Marcello Chiocchi, Armando Ugo Cavallo, Luca Pugliese, Matteo Cesareni, Daniela Pasquali, Giacomo Accardo, Vincenzo De Stasio, Luigi Spiritigliozzi, Leonardo Benelli, Francesca D’Errico, Cecilia Cerimele, Roberto Floris, Francesco Garaci, Carlo Di Donna

**Affiliations:** 1Department of Biomedicine and Prevention, University of Rome Tor Vergata, 00133 Rome, Italy; 2Division of Radiology, Istituto Dermopatico dell’Immacolata, 00167 Rome, Italy; 3Dipartimento di Scienze Mediche e Chirurgiche Avanzate, Università degli Studi della Campania Luigi Vanvitelli, 81100 Caserta, Italy; 4ASL Salerno Ds 63 Poliambulatorio Costa d’Amalfi, 84013 Salerno, Italy; 5IRCSS San Raffaele, 03043 Cassino, Italy

**Keywords:** epicardial adipose tissue, cardiac-CT, cardiac function, heart failure, epicardial adipose tissue density

## Abstract

*Background and objectives*: Epicardial adipose tissue density (EAD) has been associated with coronary arteries calcium score, a higher load of coronary artery disease (CAD) and plaque vulnerability. This effect can be related to endocrine and paracrine effect of molecules produced by epicardial adipose tissue (EAT), that may influence myocardial contractility. Using coronary computed tomography angiography (CCT) the evaluation of EAD is possible in basal scans. The aim of the study is to investigate possible associations between EAD and cardiac function. *Material and Methods*: 93 consecutive patients undergoing CCT without and with contrast medium for known or suspected coronary CAD were evaluated. EAD was measured on basal scans, at the level of the coronary ostia, the lateral free wall of the left ventricle, at the level of the cardiac apex, and at the origin of the posterior interventricular artery. Cardiac function was evaluated in post-contrast CT scans in order to calculate ejection fraction (EF), end-diastolic volume (EDV), end-systolic volume (ESV), and stroke volume (SV). *Results*: A statistically significant positive correlation between EAD and ejection fraction (r = 0.29, *p*-value < 0.01) was found. Additionally, a statistically significant negative correlation between EAD and ESV (r = −0.25, *p*-value < 0.01) was present. *Conclusion*: EAD could be considered a new risk factor associated with reduced cardiac function. The evaluation of this parameter with cardiac CT in patients with low to intermediate cardiovascular risk is possible.

## 1. Introduction

Coronary computed tomography angiography (CCT) is considered as a fundamental technique for the diagnosis of coronary artery disease (CAD) in patients with stable angina and a low-to-intermediate CAD probability [1]. Thanks to its high sensibility and specificity in detecting coronary stenosis as well as plaque instability [2], it is now part of daily clinical practice to prevent major adverse cardiac events.

In patients with CAD, despite diagnostic and therapeutic development, impaired cardiac function still has a relatively high prevalence, mostly represented by reduced left ventricle ejection fraction (EF) and left ventricle diastolic disfunction [1,2,3]. The evaluation of cardiac function is essential for diagnostic and prognostic assessment in these patients [4,5].

Although the gold standard for functional ventricle assessment is cardiac magnetic resonance (CMR) [6], this technique has several limitations, related to patient compliance and MR scanners. The development of new, more sophisticated reconstruction software related to CCT gives the possibility to assess with higher accuracy several parameters and diagnostic information, such as plaques composition, cardiac perfusion, fraction flow reserve (FFR) and even functional parameters of the left ventricle by calculating end-diastolic and end-systolic volumes (EDV and ESV), stroke volume (SV), and ejection fraction (EF) [7] giving the opportunity to radiologists and physicians to obtain a large amount of information with only one imaging exam with the lowest radiation dose possible.

In addition, epicardial adipose tissue (EAT) and its density (EAD) can be easily measured in CCT exams [8]. Several studies showed a correlation between volume and density of epicardial adipose tissue (EAT), measured after CCT, with coronary arteries calcium score, a higher load of coronary artery disease and plaque vulnerability [9,10,11,12,13].

Epicardial adipose tissue is part of the visceral adipose tissue (VAT) that surrounds the heart. It is a quantifiable, modifiable, and multifaceted tissue, located between the myocardium and the visceral pericardium and it is commonly found in atrioventricular sulci, interventricular sulci, and around the epicardial coronary arteries [14].

Epicardial adipose tissue origin from splanchnopleuric mesoderm and its cells are not only adipocytes, but even ganglia, nerves, and inflammatory, stromovascular, and immune cells.

Adipocytes located in epicardial adipose tissue are smaller than adipocytes located in other tissue such as subcutaneous adipocytes and abdominal visceral adipocytes.

Even though the primary function of epicardial adipose tissue is a protective function by providing mechanical protection, serving as an energy source to the myocardium, and producing anti-inflammatory adipokines, the excess of its lipidic content or its growth may lead pathologic mechanisms; for this reason, epicardial adipose tissue has been considered a factor that may increase the risk of metabolic syndrome (MetS).

Metabolic syndrome is complex and multi-factorial syndrome that increases cardiovascular risk.

Several definitions through the years have been proposed by the World Health Organization (WHO). The most recent worldwide harmonizing criteria were made in 2009 as a revision to the National Cholesterol Education Program (NCEP) criteria in conjunction with the International Diabetes Federation (IDF); the National Heart, Lung, and Blood Institute (NHLBI); the American Heart Association (AHA); the World Heart Federation (WHF); the International Atherosclerosis Society (IAS); and the International Association for the Study of Obesity (IASO) [15]. 

In this definition there are five established components: abdominal obesity defined by ethnicity-specific waist circumference, elevated blood pressure, impaired fasting glucose, increased triglyceride levels, and decreased high-density lipoprotein (HDL) cholesterol levels.

Development of Metabolic Syndrome (MetS) is strictly linked to pro-inflammatory action of adipocytes as a consequence of secretion of many factors including leptin and adiponectin as well as proinflammatory tumor-necrosis factor-alpha and interleukins. Inflammation leads to changes in visceral adipose tissue including activated lipolysis, release of free fatty acids, hypoxia, oxidative stress, and apoptosis of adipocytes—not only inflammation, thanks to increased secretion of TNF-alpha increases aggregation of activated macrophages from bone marrow and infiltration of macrophages into adipose tissue.

Visceral obesity or increased visceral adipose tissue (VAT), also referred to as visceral fat, is the adipose tissue that surrounds organs, as opposed to subcutaneous adipose tissue.

Increased visceral adipose tissue plays an important role in the development of metabolic syndrome, with increased cardiovascular disease, due to hypertension, heart failure, coronary artery disease (CAD), valvular disease, and arrhythmias; pulmonary disease including sleep apnea and emphysema; brain disease including stroke and dementia; various cancers; and reduced bone density.

Furthermore, visceral adipose tissue is an independent predictor of mortality in males. 

Visceral adipose tissue accumulation patients showed higher epicardial adipose tissue thickness than those with predominant peripheral fat distribution [16]. 

Epicardial adipose tissue, as one of the expressions of visceral adipose tissue, has been associated with the risk of atherosclerotic cardiovascular disease (ASCVD) including coronary artery disease. Cardiovascular disease remains the highest cause of mortality worldwide according to the World Health Organization. 

Several reports include epicardial adipose tissue as an important and central factor in the development of atrial myopathy, atrial fibrillation, thromboembolic stroke, bi-ventricular hypertrophy, and impaired bi-ventricular diastolic relaxation and filling leading to heart fail. 

Epicardial adipose tissue characteristics have also been associated with major adverse cardiac events as well as serum levels of plaque inflammatory markers [17,18,19]. In fact, epicardial adipose tissue was proven to have both paracrine and endocrine effects on other cardiac structures [20,21,22]. Particularly adipose tissue with low CT attenuation, represented by inflammatory white adipose tissue, is more likely to have some influence on the atherosclerotic process [22,23,24].

Epicardial adipose tissue has been implicated not only in the inflammatory and pro-atherosclerotic process on the coronary artery, but as demonstrated by several authors, may have an effect on cardiac pace, arrythmias, and on left and right ventricular dysfunction. In particular, Iacobellis et al. [25] and Fox et al. have demonstrated [26] that epicardial adipose tissue may reflect intra-abdominal visceral fat and could, therefore, serve as a marker of visceral adiposity. In particular, epicardial adipose tissue may be responsible for “cardiac lipotoxicity” that starts with hypertrophy and then can be followed by the development of LV dysfunction. Iacobellis et al. suggested two different mechanisms. The first one is the result of myocardial infiltration by epicardial adipocytes that move to areas between the myocardial fibers. The second one, formerly called “fatty degeneration,” is caused by deposition of triacylglycerol droplets within the cytosol of the cardiomyocytes. These two mechanisms, that may act together, can lead to a decreased metabolism of fatty acid use and increases its reliance on glucose as a fuel. This metabolic switch can be a consequence of genes such as “Peroxisome proliferator-activated receptor-α (PPARα)” [27]. Moreover, direct contact between adipose tissue and the myocardium may impact LV structure and function via paracrine secretion, especially chemokine such as Monocyte Chemo attractive Protein-1 (MCP-1) and inflammatory cytokines such as Interleukins (IL-1β, IL-6, IL-6sR) and TNF-α. This effect may be a consequence of the absence of layers between myocardium and epicardial adipose tissue. 

These effects may be one of the causes and mechanism that lead to left ventricular disfunction.

Another mechanism that can be responsible for cardiac disfunction may be the influence of epicardial adipose tissue on cardiac pace. 

Thanassoulis et al. have described an important association between epicardial adipose tissue and atrial fibrillation, the most common cardiac arrythmia. 

In particular, in the Framingham Heart Study [28] they observed that higher pericardial fat volumes were associated with a nearly 40% higher odds of prevalent atrial fibrillation.

They hypothesized that epicardial adipose tissue fat likely acts locally via mechano-structural or paracrine mechanisms.

Further observation of the Framingham Heart study was that increased epicardial adipose tissue is also associated with changes in cardiac structures and specifically increased left atrial dimensions.

The consequences of mechanic and structural changes on myocardium, especially left ventricle and atrial septum fat infiltration, may lead to electromechanical changes in atrial tissue, with consequences on atrial electric conduction, resulting in impaired cardiac function. 

Beyond mechano-structural effects, there are the effects of the secretion of chemokines and pro-inflammatory cytokines. It is known that tumor necrosis factor-alpha and interleukin 6 may have direct arrhythmogenic effects on atrial tissue and have been associated with atrial fibrillation initiation [19].

Therefore, there is an increasing interest in finding potential associations between cardiac function impairments and epicardial adipose tissue characteristics. 

The aim of the study is to look for possible associations between epicardial adipose tissue density and cardiac function in patients with low to intermediate cardiovascular risk undergoing Cardiac CT. 

## 2. Materials and Methods

In this retrospective study conducted at Policlinico “Tor Vergata”, Rome, Italy, patients provided written informed consent to use demographic, clinical, and imaging data anonymously for research purposes. The study was approved by the Internal Institutional Review Board (n.71.21).

### 2.1. Patients

The study was conducted with 93 patients with low to intermediate cardiovascular risk according to 2021 ESC Guidelines [29] and Heart Score who underwent cardiac computed tomography without and with contrast medium for patients with known or suspected coronary CAD.

### 2.2. Inclusion Criteria

Patients with low to moderate cardiovascular risk.

Patients without a history of acute coronary syndrome (ACS).

Patients without history of cardiac surgery.

Patients without history of valvular diseases.

### 2.3. Exclusion Criteria

History of coronary stents implantation or coronary by-pass surgery.

Presence at the CT scan of coronary artery disease with >50% stenosis.

Patients with valve prostheses.

Patients at high cardiovascular risk.

Patients with a history of ACS.

Presence of valvular diseases.

### 2.4. Coronary CT

Coronary CT was performed using a 512-slice CT (GE-Healthcare CT Revolution System, General Electric, Milwaukee, WI, USA) using retrospective gating in ECG-activated high-angle spiral acquisition mode. All images were transferred to an external workstation (ADW-4.7; GE Healthcare) for post-processing analysis. After non contrast CT acquisition, a non-ionic iso-osmolar contrast agent (Iomeron 400 mg/mL-Bracco Imaging S.p.A.-Italy) was injected into an antecubital vein through a 20-gauge catheter using a double-short injector (Nemoto Kyorindo, Tokyo, Japan). 

### 2.5. Epicardial Adipose Tissue Density

Left ventricular function was evaluated by a dedicated workstation using the CardiqXpress software in post-contrast acquisitions, whereas epicardial adipose tissue density was measured on non-contrast scans using a dedicated workstation (Advantage Workstation-4,7(ADW); GE Healthcare, General Electric, Boston, MA, USA).

EAT density was measured on basal scans, using a 4-chamber projection, at the level of the proximal left and right coronary artery, lateral free wall of both ventricles, at the origin of the posterior interventricular artery, and at the level of the cardiac apex. (Figure 1A–F).

### 2.6. Cardiac Function

Left ventricle function has been evaluated using post-contrast acquisition evaluated with Cardiq Xpress, a dedicated software included in the workstation used (ADW-4.7; GE-Healthcare) by two radiologists in consensus with respectively, 5 and 3.5 years of experience in cardiac imaging (Figure 2).

### 2.7. Statistical Analysis

The statistical analysis was performed with R V 3.4.4 [R Core Team (2018). R: A language and environment for statistical computing. R Foundation for Statistical Computing, Wien, Austria] [30].

A Pearson test was used to assess the correlation between variables. A *p*-value < 0.05 was considered statistically significant. 

The correlation coefficient was considered negligible if between 0 and 0.19, weak if between 0.2 and 0.29, moderate if between 0.3 and 0.49, strong if between 0.5 and 0.79, and very strong if between 0.8 and 1.

Linear regression analysis was performed when the correlation with epicardial adipose tissue density was statistically significant.

## 3. Results

Of 93 patients, 25 were female (26.95%) and the BMI was 27.9 ± 6.3 Kg/cm. 

Clinical characteristics were the following: age 66.4 ± 10.01 years, sex (F) 25 (26.9%), smokers were 9 (9.7%), patients suffering arterial hypertension were 19 (21%), patients with diabetes 17 (18.3%), patients with dyslipidemia 45 (48.4%), patients with a history of acute coronary syndrome were 0, serum creatinine levels 0.94 ± 0.18 mg/dL, blood urea nitrogen levels were 13.45 ± 4.29 mg/dL, systolic blood pressure was 125.5 ± 3.8 mmHg, total cholesterol levels were 152 ± 6.9 mg/dL, low density lipoprotein level was 90 ± 6.5 mg/dL, high density lipoprotein level was 64 ± 6 mg/dL, and total Heart Score was 7.5 ± 4.8%.

Demographic and clinical data are shown in Table 1.

Correlation between variables are showed in the correlogram (Figure 3).

A positive, weak correlation between epicardial adipose tissue density and ejection fraction (r = 0.29, r^2^ = 0.08, *p*-value < 0.01) (Figure 4) and a negative weak correlation between epicardial adipose tissue density and End Systolic Volume (ESV, r = −0.25, r^2^ = 0.08, *p*-value < 0.01) were found (Figure 5). 

Correlation between epicardial adipose tissue density and End Diastolic volume was negligible (r = −0.18) and not statistically significant (*p* = 0.08). The correlation between epicardial adipose tissue density and Stroke Volume was negligible (r = −0.0.1) and not statistically significant (*p* = 0.9). 

Correlation values between epicardial adipose tissue density and cardiac function parameters are summarized in Table 2.

## 4. Discussion

To our knowledge this is the first study investigating the role of epicardial adipose tissue and its density as possible cardiovascular marker of risk.

There are several studies that investigate the role of epicardial adipose tissue with inflammation and plaques vulnerability that may increase cardiovascular risk. In particular, Goeller et al. and Abazid et al. described a correlation between epicardial adipose tissue density, coronary calcification, and subclinical prevalence of coronary artery disease.

White adipose epicardial tissue with low density is related to higher BMI and higher coronary calcium score [17,27,28,29,30,31,32,33,34,35,36,37], thus suggesting the presence of chronic inflammation determined by white epicardial adipose tissue that may increase cardiovascular risk.

Goeller et al. demonstrate an epicardial adipose tissue density significantly lower in subjects with coronary calcium or a coronary calcium score ≥ 100 compared to subjects with a coronary calcium score of 0 [17].

Franssens et al. also described an association between lower epicardial adipose tissue density and greater extent of coronary artery calcification too. 

In these studies, lower epicardial adipose tissue density measured on a basal CT scan has been associated with higher BMI and a higher coronary calcium score, as already described for density of visceral adipose tissue [31,32], suggesting an increased cardiovascular risk.

It is demonstrated that adipose tissue with lower CT attenuation is related to adipocyte hypertrophy and hyperplasia following excess lipid accumulation in diet-induced obesity and insulin resistance [23,31,33,34,35]. This adipose tissue is linked to metabolic syndrome, and it is associated with systemic low-grade inflammation because it produces different molecules, such as Plasminogen activator inhibitor-1 (PAI-1) and Monocyte chemoattractant protein 1 (MCP-1), that are correlated to increased risk of vascular inflammation and progression of atherosclerosis.

Moreover, white low density EAT, as demonstrated by Goeller et al., is associated with the reduction level of Adiponectin. A low adiponectin expression has been reported to be associated with acute coronary syndrome and progression of atherosclerosis because of its anti-inflammatory and anti-atherogenic effects.

Antonopolus et al. investigated the role of perivascular adipose tissue (PVAT) and developed an alternative metric called the perivascular CT fat attenuation index (FAI) [20].

In this study, authors suggested that adipocytes in PVAT respond to proatherogenic processes in the underlying vascular wall that modify their biology and assume that inflammatory signals from the human arterial wall may diffuse into the PVAT to influence adipocyte lipid content by affecting biological processes such as adipocyte differentiation, proliferation, and lipolysis [36].

They used CT attenuation of adipose tissue because densitometric values reflect the balance between lipid and aqueous content, and they affirmed that EAT density may be considered a marker of adipocyte size/adipose tissue lipid content.

Several studies found an important role of systemic chronic inflammation in non-communicable diseases, such as cardiovascular disease. Conditions such as obesity, hypertension, autoimmunity, and aging induce a series of inflammatory patterns that have been linked to heart failure. Inflammatory byproducts promote coronary microvascular endothelial inflammation, oxidative stress, lowering nitric oxide and cardiomyocytes loss, but they also have a direct effect on cardiac resident immune cells thus perpetrating local chronic inflammation. One of the inflammatory patterns that has been identified is the activation of NLRP3 inflammasome, which leads to IL-1-β and IL-18 secretion. IL-1-β decreases the expression of SERCA (sarcoplasmic reticulum calcium ATPase), and phospholamban (PLB), which can induce diastolic dysfunction, while IL-18 induces fibrosis and cardiac hypertrophy resulting in diastolic stiffness and concentric remodeling. Therefore, the correlation between chronic inflammation and heart failure suggests the possibility of using the mediators of this patterns as possible clinical targets [37]. 

Milanese et al. [38] showed that epicardial adipose tissue is increased in diabetic patients compared with a non-diabetic population and they suggested that the increased volume of epicardial adipose tissue might increase cardiovascular risk of these patients because epicardial adipose tissue is able to release systemic mediators of endothelial stress and enhance coronary vasculopathy by means of increased paracrine metabolic activity.

They even reported a systematic review from Spearman et al. [39] where the authors affirmed that epicardial adipose tissue surrounding coronary arteries could be a factor able to promote atherosclerosis, arterial stiffness and coronary artery calcification, even though the biochemical and biomolecular mechanisms are not fully understood yet.

Moreover, Keiler et al. [40] affirmed that Sub venous Epicardial Adipose Tissue (SEAT)—which acts as an electrical insulation—and its properties, such as density and thickness, may be relevant in prognostic evaluation of cardiac resynchronization in patients with heart failure.

Van Woerden et al. [41] affirmed that epicardial adipose tissue has been linked to biomarkers of myocardial damage, ventricular hypertrophy, increased cardiac filling pressures, and worse exercise capacity that lead to heart failure. Even though their study was based on cardiac magnetic resonance (CMR), it demonstrates that patients with a high representation of epicardial adipose tissue had a significantly higher relative event rate of cardiovascular major events compared to patients with low epicardial adipose tissue, supporting the concept that epicardial adipose tissue plays an active and important role in the pathophysiology of heart failure. Because of a pro-inflammatory function of epicardial adipose tissue, enabling it to produce, adipokines and cytokines cause microvascular disfunction, beyond mechanical compression and constriction, thus, leading to reduced heart function or heart failure. Moreover, they affirmed that was positively associated with increased pulmonary artery systolic pressure. These findings, in line with previous data, showed an association between epicardial adipose tissue and right-sided filling pressures and may suggest that an abundance of epicardial adipose tissue surrounding the heart may lead to right heart overload.

### 4.1. Cardiac-CT Role

Cardiac computed tomography is gaining attention worldwide for several reasons: the technological improvements are leading to a new generation of scanners with the possibility of exposing patients, in very reduced times, to low radiations dose while obtaining very high (up to sub millimetrical) resolution images, useful for the representation of very small structures, such as coronary arteries. 

In recent years a robust evidence base for the use of cardiac CT in diagnoses of heart disease has been developed, in addition to prognostication and therapy modulation (both medical and interventional).

It is not surprising that several guidelines [41,42,43] started to incorporate cardiac computed tomography as a valuable test to investigate coronary artery disease risk in stable patients with low to intermediate cardiovascular risk factors. Moreover, cardiac computed tomography can be performed in patients with previous severe coronary artery disease in order to evaluate stenting therapy or by-pass surgery.

With our analysis we show that quantification of epicardial adipose tissue in cardiac computed tomography is possible and relatively easy for physicians. 

The lower the adipose density in measured Hounsfield Units, the higher the lipid content of adipocytes in epicardial fat. The positive, weak correlation with ejection fraction and the negative weak correlation with end systolic volume may explain a potential role of epicardial adipose tissue in influencing cardiac function—the real mechanism in not completely explained by our results. 

There is the possibility that the endocrine-paracrine effects of epicardial adipose tissue may directly influence myocardial contractility and myocardial pacing. Additionally, epicardial adipose tissue could contribute to the atherogenic process, playing a role in the determination of coronary artery disease or microcirculation damage, thus leading to myocardial impaired contractility. 

In our experience we did not include in the analysis patients with coronary artery disease, so the first mechanism looks more probably related to the findings. Anyway, the aim of our study was not the investigation of these mechanisms, so further investigation is needed to fully understand the effects of epicardial adipose tissue on myocardial contractility.

Interestingly, no statistically significant correlation between epicardial adipose tissue density was found with end diastolic volume and stroke volume. The explanation of this result is not clear. There is a possibility that epicardial adipose tissue may influence myocardial contractility and not myocardial relaxation during diastole, but this is only a speculative explanation. Additional evidence is needed to fully understand and explain this result.

Nowadays, thanks to technological improvement in computed tomography scans, with faster acquisition time, low radiation dose, and increased availability of modern computed tomography scans, it is possible to extract from a single computed tomography study several information about the heart, coronary arteries, myocardial tissue, and about several structures of mediastinum and chest with only a single scan.

This important achievement may help physicians to tailorize therapies for each patient based on CT reports, using drugs already used to reduce visceral fat deposit, such as GLP-1 (glucagon-like peptide 1) receptor agonists and SGLT-2 (sodium-glucose co-transporter 2) inhibitors.

Therefore, cardiac computed tomography might become a comprehensive methodology for cardiac analysis, assessing simultaneously the presence of coronary artery disease, cardiac function, and the presence of imaging biomarkers of cardiac disease, such as epicardial adipose tissue density or other related features.

### 4.2. Limitations

This study has several limitations: (1) the small sample size which is slightly unbalanced for sex (there is a prevalence of males: 73.1%); (2) the single center analysis; (3)the retrospective design; (4) cardiac CT scans were acquired with only one scanner, thus reducing generalizability of results; (5) cardiac function analysis was performed only in cardiac CT images, without an additional, gold standard methodology used for comparison(e.g., echocardiography or cardiac MRI); (6) the analysis was performed only in patients with low to moderate cardiovascular risk, without including patients at high risk for CAD or with known coronary artery disease; (7) inter reader agreement analysis for both epicardial adipose tissue and cardiac function assessment was not performed.

## 5. Conclusions

To our knowledge this is the first study that investigated the association of epicardial adipose tissue and its density with cardiac function evaluated during cardiac CT scans, to obtain, from a single diagnostic investigation, information about cardiac function. 

Epicardial adipose tissue, because of its combined pro-inflammatory and pro-arrythmogenic effects and because of the mechano-structural changes affecting the atrial septum and left ventricle myocardium, may lead to abnormalities of electric conduction, cardiomyocyte’s contractile function and may also influence gene expression through receptors on cardiomyocytes. 

For these several reasons, epicardial adipose tissue could be considered a new imaging prognostic factor of cardiac failure, that could be mentioned in radiological reports with the evaluation of cardiac function. 

Even Artificial Intelligence (AI) algorithms may play an important role in the analysis of epicardial adipose tissue. Body composition imaging is a novel concept based on quantitative analysis of body tissues. Machine learning is a group of techniques that extrapolate or classify data through complex mathematical models [44]. 

In this scenario, these algorithms may offer robust and feasible solutions to quantify adipose tissue and might be applied to epicardial adipose tissue density quantification.

Moreover, “omics” sciences such as radiomics and metabolomics may offer interesting and additional integrated approaches in order to evaluate epicardial adipose tissue composition in biomedical images. Additionally, by the analysis of data extracted at minimal or no additional cost to healthcare systems, these techniques may help in the definition of new diagnostic and prognostic models for cardiac disease.

Thus, the epicardial adipose tissue density intended as a cardiac pathology biomarker, associate to new technology in the cardiac computed tomography and cardiac magnetic resonance spectrum may play an important role in the identification of new potential therapeutic and/or preventive strategies to prevent and to reduce cardiovascular risk and reduce major cardiovascular events.

Further analysis will be needed to prove the potential of epicardial adipose tissue density measured in cardiac CT as a biomarker, even in addition to new technologies such as radiomic, metabolomic analysis, and artificial intelligence algorithms. However, our experience may lay ground for future investigations in this scientific field.

## Figures and Tables

**Figure 1 medicina-59-00232-f001:**
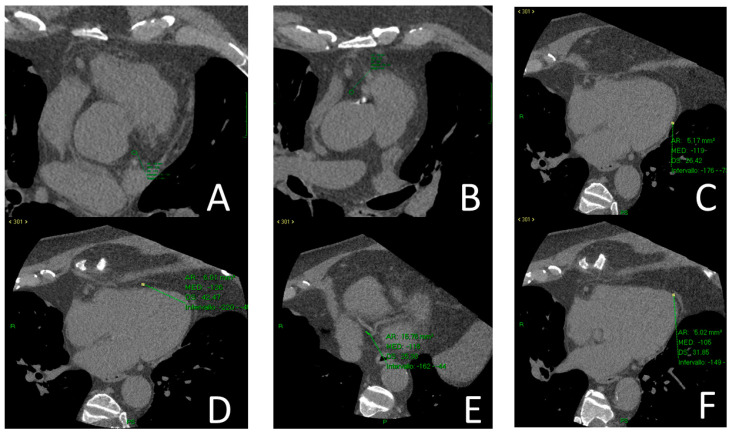
Epicardial Adipose Density measure using ROI at proximal left coronary artery (**A**), proximal right coronary artery (**B**), the lateral free wall of the left ventricle (**C**), the lateral free wall of the right ventricle (**D**), at the origin of the posterior interventricular artery (**D**), and at the level of the cardiac apex (**F**).

**Figure 2 medicina-59-00232-f002:**
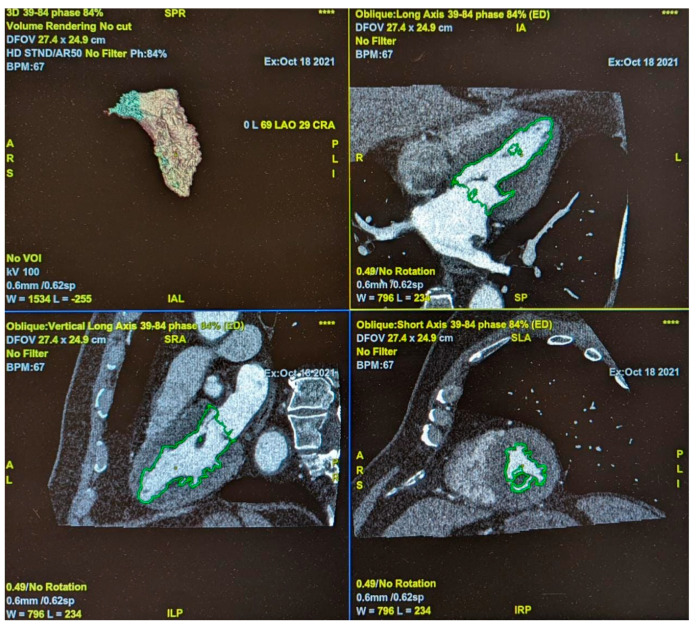
Evaluation of Left Ventricle function using the ADW-4.7 workstation (GE Healthcare).

**Figure 3 medicina-59-00232-f003:**
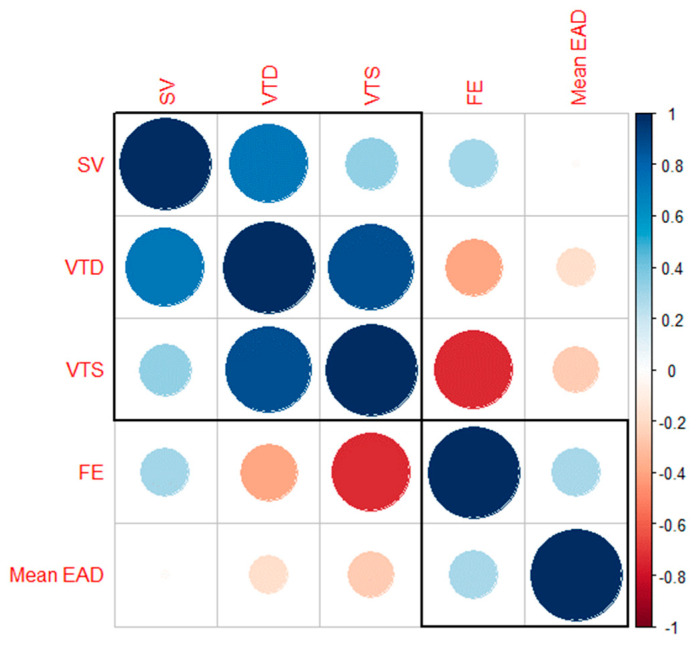
Correlogram showing correlation between variables.

**Figure 4 medicina-59-00232-f004:**
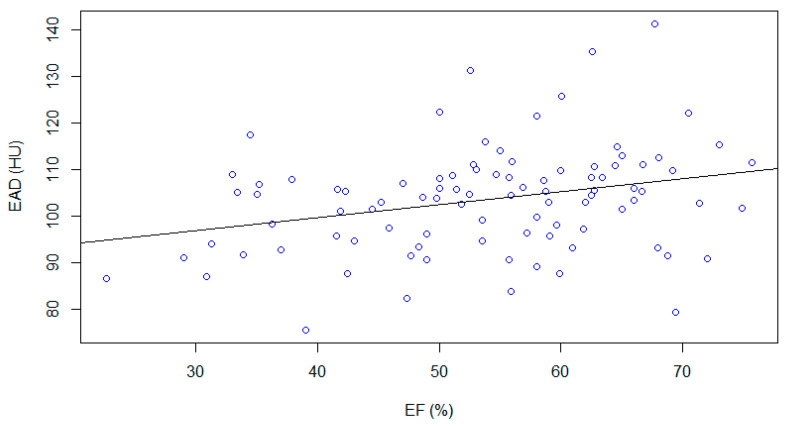
EF values plotted with EAD. A positive, weak correlation was found (r = 0.29, r^2^ = 0.08 *p*-value < 0.01).

**Figure 5 medicina-59-00232-f005:**
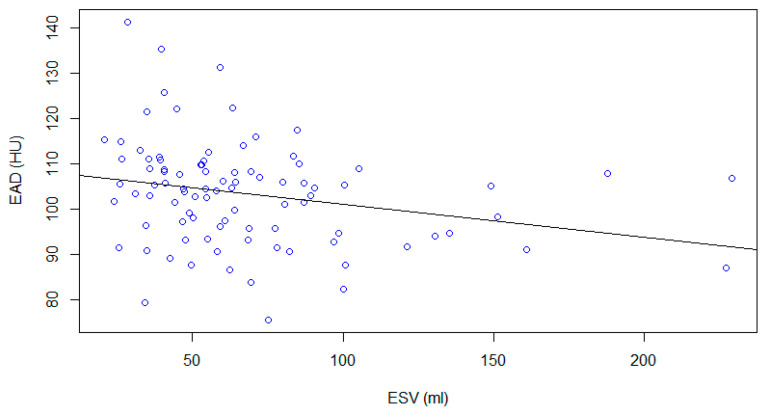
ESV values plotted with EAD. A negative, weak correlation was found (r = −0.25, r^2^ = 0.08, *p*-value < 0.01).

**Table 1 medicina-59-00232-t001:** Characteristics of the study population (93 patients). Data are reported as mean ± SD for continuous variable and as absolute value and percentage for categorical variables.

**Age (years)**	66.4 ± 10.01
**Sex (F)**	25 (26.9%)
**BMI (Kg/cm)**	27.9 ± 6.3
**Smoke**	9 (9.7%)
**Hypertension**	19 (21%)
**Diabetes**	17 (18.3%)
**Dyslipidemia**	45 (48.4%)
**ACS**	0
**Serum creatine(mg/dL)**	0.94 ± 0.18
**BUN (mg/dL)**	13.45 ± 4.29
**Systolic Blood Pressure (mmHg)**	125.5 ± 3.8
**Total cholesterol (mg/dl)**	152 ± 6.9
**HDL (mg/dL)**	64 ± 6
**LDL (mg/dL)**	90 ± 6.5
**Heart Score (%)**	7.5 ± 4.8

Mean epicardial adipose tissue density was 103 UH ± 11.46. Ejection fraction was 53.83 ± 11.88 (%); End Diastolic Volume was 139.13 ± 32.03 mL; End Systolic Volume was 67.68 ± 40.01 mL; Stroke Volume was 72.65 ± 23.17 mL.

**Table 2 medicina-59-00232-t002:** Correlation between EAD and Cardiac Function parameters. Person test results (r) and *p* values are also reported.

	r	*p*
**Ejection fraction (%)**	0.2885017	0.00504
**End Diastolic Volume(mL)**	−0.1793627	0.08538
**End Systolic Volume (mL)**	−0.2537976	0.0141
**Stroke Volume (mL)**	−0.01243054	0.9059

## Data Availability

The data sets generated and analyzed during the current study are available from the corresponding author on reasonable request.

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
