# Peer review of "Cardiac Computed Tomography Evaluation of Association of Left Ventricle Disfunction and Epicardial Adipose Tissue Density in Patients with Low to Intermediate Cardiovascular Risk"

_medicina, 2023, doi:10.3390/medicina59020232_

Round 1
Reviewer 1 Report
The paper brings original and interesting data on the association between EAT and cardiac function.
Improvements to the manuscript may include:
- use of term ‘marker of risk’ instead of ‘risk marker
- improve patient characterization, with clinical description of their cardiac status and score evaluation of thir risk;
- include data on calcium score, if available.
Author Response
The paper brings original and interesting data on the association between EAT and cardiac function.
Improvements to the manuscript may include:
- use of term ‘marker of risk’ instead of ‘risk marker
Answer: Thank you, we have used the terminology suggested where applicable.
- improve patient characterization, with clinical description of their cardiac status and score evaluation of thir risk;
- include data on calcium score, if available.
Answer: Thank you for this comment. In Table 1 we have added clinical information needed to calculate the Heart Score and the mean Heart Score of patients included in the analysis. Unfortunately, we didn’t collect data about calcium score.

Reviewer 2 Report
This is an outstanding article evaluating the use of epicardial adipose tissue as a marker for cardiac function assessment. The sound scientific methods (although done retrospectively) used and clear, to the point, description make this article easy to read and very significant for clinical use.
Have you assessed the development of the arrhythmias for these patients (maybe not part of this work)?
Please edit Figure 2 (software window is not necessary) and there is an extra "enter" in the line 152 (insulin resistance)
Author Response
This is an outstanding article evaluating the use of epicardial adipose tissue as a marker for cardiac function assessment. The sound scientific methods (although done retrospectively) used and clear, to the point, description make this article easy to read and very significant for clinical use.
Have you assessed the development of the arrhythmias for these patients (maybe not part of this work)?
Answer: Thank you for this observation. We did not evaluate the development of arrythmias for this work. This could be of potential interest for future investigations.
Please edit Figure 2 (software window is not necessary) and there is an extra "enter" in the line 152 (insulin resistance)
Answer: Thank you, we have modified the manuscript accordingly.
